# Individual Differences in Adolescents’ Civic Engagement: The Role of Civic Discussions with Parents and Environmental Sensitivity

**DOI:** 10.3390/ijerph20136315

**Published:** 2023-07-07

**Authors:** Giusy Danila Valenti, Alida Lo Coco, Nicolò Maria Iannello, Cristiano Inguglia, Michael Pluess, Francesca Lionetti, Sonia Ingoglia

**Affiliations:** 1Department of Psychology, Educational Science and Human Movement, University of Palermo, 90128 Palermo, Italy; giusydanila.valenti@unipa.it (G.D.V.); alida.lococo@unipa.it (A.L.C.); sonia.ingoglia@unipa.it (S.I.); 2Department of Law, University of Palermo, 90128 Palermo, Italy; nicolomaria.iannello@unipa.it; 3Department of Biological and Experimental Psychology, Queen Mary University of London, London E1 4NS, UK; m.pluess@qmul.ac.uk; 4Department of Neurosciences, Imaging and Clinical Sciences, University G. D’Annunzio Chieti-Pescara, 66100 Chieti, Italy; francesca.lionetti@unich.it

**Keywords:** civic engagement, adolescence, environmental sensitivity, parent-child relationship

## Abstract

The main goal of the current study was to examine the direct and moderating effects of civic discussions with parents and environmental sensitivity using both the total score and its specific dimensions (i.e., Aesthetic Sensitivity, AES; Ease of Excitation, EOE; Low Sensitivity Threshold, LST) on youth civic engagement (attitudes and behaviours). The empirical analysis relied on a questionnaire-based survey conducted on a sample of 438 adolescents (30% males), aged between 14 and 18 years (*M* = 16.50, *SD* = 1.36). We used a structural equation model (SEM) with latent variables and the latent moderated structural equation (LMS) method to test our hypotheses. Our results showed that civic discussions with parents were positively and significantly associated with general environmental sensitivity and with AES and predicted both civic attitudes and civic behaviours; EOE was negatively and significantly related to civic behaviours; AES was positively and significantly related to civic attitudes; and LST was not significantly related to either civic attitudes or behaviours. Contrary to our expectations, environmental sensitivity did not moderate the relationship between civic discussions with parents and civic engagement. Our study further highlights the relevance that parents have in shaping their children’s civic engagement and makes a novel contribution regarding how differences in perceiving and processing environmental stimuli can affect beliefs and behaviours toward community issues among young individuals.

## 1. Introduction

Modern society is characterized by political, economic, and social changes that have led individuals—especially the younger ones—to reduce their level of commitment to their community and to be less connected to public affairs [1]. As also reported by some scholars [2,3], these changes have modified and reduced the way people form attachments to their own large community, since issues are regarded as private rather than collective. However, because commitment to the community is considered a fundamental prerequisite to the development of democracy, it is important to understand which factors may promote its occurrence.

In response to the urgent need for knowing how youths develop a sense of involvement for their communities, there has been an increase in interest on civic engagement [4], defined as individual and collective actions meant to identify and address issues of public concerns [5]. Scholars have investigated the psychosocial dimensions associated with the development of civic engagement in adolescence, taking into account both contextual factors, such as exposure to social media, and parents’ civic and political participation, and individual or dispositional factors, such as common personality traits [6,7,8].

However, most of these studies have focused on contextual factors, whereas little emphasis has been placed on individual ones or on their interplay in influencing adolescents’ civic engagement [9]. With the current contribution, we propose that identifying how both contextual and dispositional factors, and their interplay, are associated with civic engagement in adolescence may contribute to a deeper comprehension of this phenomenon, which can be intended as a preferential arena in which to observe the complex individual-environment transactions [9]. Thus, the current study aims to examine how contextual factors, such as discussions with parents about community issues, and dispositional factors, such as the temperament trait of environmental sensitivity, are associated with adolescents’ civic engagement.

### 1.1. Civic Engagement: Conceptual Frameworks

According to the American Psychological Association [5], civic engagement refers to the individual and collective actions aimed at identifying and addressing issues of public concern, including voluntarism, participation in political parties, working with other people, or interacting with institutions in order to find solutions to problems of the community. From the perspective of developmental psychology, the emergence and the maintenance of civic engagement are considered core features of youth’s positive development, and studies focused on factors that may promote and enhance individuals’ contributions to civil society have received much attention [10,11]. Within the positive youth development model [12,13], civic engagement is thought of as the result of mutually beneficial relations—termed *adaptive regulations*—between individuals and their developmental context. It reflects the idea that when people support and contribute to the support of social institutions, people will in turn experience opportunities to thrive and flourish. As Lerner et al. [14] have underlined, “there exists an adaptive social ‘contract’: Individuals act to support a social world that, in turn, acts to support the individual, qua individual” (p. 150). In other words, through the positive bidirectional regulations between individuals and their context, a social adaptive contract is set up that allows and optimizes the functioning of both components.

Doolittle and Faul [3], in an effort to draw up a valid tool to measure civic engagement, have underscored that civic engagement, far from being a unidimensional construct, consists of two dimensions: attitudes and behaviours. In particular, *civic attitudes* refer to the personal beliefs and feelings that people have regarding their commitment in their own community and their perceived ability to make a difference in it, whereas *civic behaviours* are conceptualized as the actions that people actually perform to engage with and make a difference in their community.

### 1.2. Contextual Factors and Youth Civic Engagement: The Role of Discussions with Parents

Most research on civic engagement has tried to identify the contextual resources that are associated with higher civic participation of various kinds, such as volunteering, and individual and group efforts to solve community problems [15,16,17,18]. Several authors have investigated the role of parents on youth civic engagement development, highlighting their importance in shaping children’s civic attitudes and behaviours. In particular, some scholars have found that parents’ participation in civic and political activities is fundamental for the development of youth civic engagement [19,20,21]. Akin to this line of research, our study aimed at exploring the relation between adolescents’ civic engagement and their discussions with parents about civic issues.

From this perspective, there is empirical evidence that youths who are used to discussing political and civic issues with their parents tend to show higher levels of civic engagement in terms of volunteering and political activism [11,17,21,22,23,24,25]. These kinds of discussions support youths in formulating their own views on these issues, stimulating them to actively seeking information on civic matters, and push them to perform civic behaviours [17,26,27].

### 1.3. Environmental Sensitivity: Definition and Conceptualization

People may differ in their response to contextual factors as a function of dispositional factors, such as *environmental sensitivity*, the ability to perceive and process environmental stimuli [28]. Individuals seem to differ with regard to the ways they register, process, and respond to the same environmental conditions, with some generally being more and others less sensitive to external stimuli [28]. Environmental sensitivity is influenced by genetic, psychological, and contextual factors [29,30,31,32], and it is understood to be a relatively stable temperament trait that appears in childhood and that is shaped further by contextual conditions.

Although Aron and Aron [33] originally hypothesized that sensitivity should be considered a unidimensional construct, some studies [34,35,36] demonstrated the existence of three different factors, offering a deeper understanding of the concept. The three factors that emerged from these studies are (1) Aesthetic Sensitivity (AES), which captures the response to aesthetic stimuli (such as being sensitive to arts and music); (2) Low Sensory Threshold (LST), which describes uncomfortable sensory arousal to external stimuli (such as response to bright lights and loud noises); and (3) Ease of Excitation (EOE), which reflects being easily overwhelmed by external and internal needs (such as negative response to having too much to do). The three factors reflect different facets of the construct and show specific patterns of associations with other personality traits, as reported in previous studies [34,35,37]. For instance, neuroticism has been found to be associated with EOE and LST, whereas agreeableness and openness have been found to be related to AES [36,38,39,40]. Also, Pluess et al. [34] and Lionetti et al. [36] indicated that a bifactor solution, including a shared general factor in addition to the three separate factors, showed a good fit to their data, suggesting the use of the total score of the scale.

### 1.4. A Plausible Linkage between Environmental Sensitivity and Civic Engagement

Previous studies reported that environmental sensitivity was directly associated with several psychological indicators of well-being [41,42] and functioned as a significant moderator. Specifically, with regard to the moderating role, the Vantage Sensitivity framework [43,44] points out that more sensitive individuals are more likely to benefit from positive environmental influences. For instance, they are more responsive to features of the environment that promote well-being, such as sensitive parenting; high-quality childcare; supportive friendships; positive life events; or other forms of support, like psychological interventions or psychotherapy. These studies provided some empirical evidence that environmental sensitivity moderates the effects of interventions, treatments, and parenting practices on children and adolescents’ psychological adaptation [44,45,46,47].

Consequently, it can be argued that youths with higher levels of environmental sensitivity will be more susceptible to the developmental contexts provided by parents (i.e., discussions at home), teachers (i.e., citizenship education), or public institutions (i.e., social advertising), potentially encouraging civic engagement. To the best of our knowledge, there are no studies that have investigated the associations between environmental sensitivity and civic engagement. Thus, a better understanding of this relationship is required to improve our comprehension of youth civic engagement and offers new insights into predictors of positive attitudes and behaviours toward community issues. In light of these arguments, the current study tries to fill this gap in the literature by exploring whether environmental sensitivity has a direct influence on civic engagement and whether it functions as a moderator in the relationship between civic discussions with parents and civic engagement.

### 1.5. Gender Differences in Environmental Sensitivity and Civic Engagement

The literature highlights gender differences in both environmental sensitivity and civic engagement. With regard to environmental sensitivity, previous studies reported that females scored significantly higher than males, although these differences are not substantial [34,48,49]. This is not surprising considering that females usually show higher scores in neuroticism, agreeableness, and openness than males [50,51], and these personality traits are often related to environmental sensitivity. Regarding civic engagement, it seems that men are more likely than women to show a higher propensity in being engaged in political activities, such as membership to political parties, involvement in political campaigns, or attendance at political meetings [52,53]. Women, on the other hand, seem to be more inclined to take part in non-institutionalized forms of participation, such as boycotting, signing a petition, or donating [52,54,55]. In addition, some studies report that females, compared with males, are more involved in non-political volunteering, community service, and local civic organization [6,56,57].

### 1.6. Aims and Hypotheses

Based on theory and existing empirical studies, the general goal of the present study was to examine how adolescents’ civic engagement is associated with a dispositional characteristic (environmental sensitivity) and a contextual factor (discussions with parents about civic issues), as well as the interplay between both. Our first goal was to investigate whether both environmental sensitivity (total score and the three subdimensions) and civic discussions with parents are associated with adolescents’ civic engagement (intended both as attitudes and as behaviours). Specifically, we hypothesized positive associations between environmental sensitivity and civic engagement, arguing that individuals who are more sensitive to their surroundings would be more likely interested in issues affecting their community, have positive beliefs about their commitment in community, and be involved in civic activities. Moreover, we hypothesized positive associations between civic discussions with parents and civic engagement, on the basis that youths who are more frequently involved in discussions with parents about civic issues are more likely to have positive beliefs about their commitment to community and are more involved in civic activities. Additionally, we investigated the associations between environmental sensitivity and civic discussions with parents. Fully aware of the lack of studies supporting the associations between these two constructs, we examined such a relationship in an exploratory manner.

The second goal of the study was to test whether adolescents’ environmental sensitivity functions as a moderator in the relationship between civic discussions with parents and civic engagement. According to the Vantage Sensitivity framework [43,44], we expected a higher magnitude of this association in adolescents with higher levels of environmental sensitivity. Previous studies reported that individuals tend to respond in a different manner to positive external stimuli as a function of their level of environmental sensitivity [47,58]. Although these studies considered environmental sensitivity as a global construct, we also investigated whether the three subdimensions functioned as significant moderators. Notwithstanding, we did not have specific hypotheses regarding the inclusion of each specific facet as moderators; we investigated such associations in an exploratory manner.

Finally, we investigated gender differences in the study variables. Precisely, regarding environmental sensitivity, we expected that females would report higher scores in the total score and all three subdimensions. Concerning civic engagement, we hypothesized that higher levels of civic engagement (both attitudes and behaviours) would be found in females than in males because in the current study, we did not include items investigating political participation or institutional civic involvement.

## 2. Materials and Methods

### 2.1. Participants

The sample was originally composed of 448 Italian adolescents living in Sicily, Southern Italy. After the deletion of 2 cases with missing data on gender and 8 multivariate outliers on study variables, the final sample included 438 participants (30% males), aged between 14 and 18 years (*M* = 16.50, *SD* = 1.36). The participants attended a high school located in Palermo and they were asked to indicate their year of attendance in high school: 15% were enrolled in their first year, 14% were enrolled in their second year, 22% were enrolled in their third year, 28% were enrolled in their fourth year, and 21% were enrolled in their fifth year. With regard to parents’ educational level, 44% of fathers had a middle school diploma or lower (8 years of education or less), 39% had a high school diploma (13 years of education), and 17% had a university degree or higher (18 years of education or more); regarding mothers, 38% had a middle school diploma or lower (8 years of education or less), 41% had a high school diploma (13 years of education), and 21% graduated from university (18 years of education or more). With regard to parent’s occupational status, 92% of fathers were employed, 5% were unemployed, and 3% were retired; 65% of mothers were employed, 34% were unemployed, and 1% were retired. Most parents were married (88%), 10% were separated or divorced, and 1% were partnered.

### 2.2. Procedure

Data were collected through an online survey distributed in a high school in Palermo after receiving consent from the school. The inclusion criteria to be part of the study were to be between 14 and 18 years old and to be Italian speakers. All procedures were performed in compliance with the Declaration of Helsinki regarding research on human participants and approved by the Internal Ethic Committee of the University of Palermo (protocol code 87/22). Written informed consent was obtained from all adolescents or from minors’ parents.

### 2.3. Measures

#### 2.3.1. Civic Engagement

Adolescents were administered the Civic Engagement Scale (CES; [3]), a 14-item scale aimed at assessing civic attitudes (8 items; e.g., “I feel responsible for my community”) and behaviours (6 items; e.g., “I help members of my community”). Items were rated on a 5-point scale ranging from 1 (*completely disagree*) to 5 (*completely agree*) for the Civic Attitudes subscale and from 1 (*never*) to 5 (*always*) for the Civic Behaviours subscale. The internal consistency was good: Cronbach’s α was 0.81 for each subscale. In order to define civic attitudes and civic behaviours as latent variables in the tested models, the technique of parcelling was used to derive a smaller number of observed indicators. Both factors were measured using three parcels computed as the mean score of 2 to 3 items. To test the factorial validity of the model, we conducted a confirmatory factor analysis (CFA; see the Data Analyses section for model fit criteria) based on the Maximum Likelihood (ML) estimation procedure that supported a 2-factor model, χ^2^(8) = 21.70, *p* < 0.01, CFI = 0.972, RMSEA = 0.063, and SRMR = 0.027. The reliability coefficients were good: factor determinacies based on the CFA model [59] were 0.84 for civic attitudes and 0.88 for civic behaviours.

#### 2.3.2. Environmental Sensitivity

Adolescents were administered the Highly Sensitive Child Scale (HSC; [34], Italian version by Nocentini et al. [47]), the most popular scale developed for assessing environmental sensitivity. It is a 12-item scale containing three subscales: (a) *Ease of Excitation* (EOE), which refers to being easily overwhelmed by external and internal demands (5 items; e.g., “I find unpleasant to have a lot going on at once”); (b) *Aesthetic Sensitivity* (AES), which captures the response and appreciation of aesthetic stimuli (4 items; e.g., “Some music can make me really happy”); and (c) *Low Sensitivity Threshold* (LST), which reflects unpleasant arousal to external stimuli (3 items; e.g., “I don’t like loud noises”). Items were rated on a 5-point scale ranging from 1 (*not at all*) to 5 (*extremely*). To test the factorial validity of the scale, we conducted a CFA based on ML estimation procedure. The 3-factor model did not fit the data well, χ^2^(51) = 145.49, *p* < 0.001, CFI = 0.886, RMSEA = 0.065, and SRMR = 0.049. Consequently, it was modified by deleting item 7 (“I don’t like watching TV programs that have a lot of violence in them”), which had a non-significant factor loading, and by adding the residual covariances between items 6 (“I am annoyed when people try to get me to do too many things at once”) and 8 (“I find it unpleasant to have a lot going on at once”). The modified model had a good fit to the data, χ^2^(40) = 90.85, *p* < 0.001, CFI = 0.925, RMSEA = 0.054, and SRMR = 0.047. The internal consistency of the scale was good: factor determinacies were 0.85, 0.86, and 0.84 for EOE, AES, and LST, respectively, and Cronbach’s α were 0.71, 0.67, and 0.64, for EOE, AES, and LST, respectively. Although the internal consistency of each subscale was not optimal, these results were in line with that reported in the original validation study and may also be explained by the low number of items in each scale capturing the different aspects of environmental sensitivity. Cronbach’s α for the total scale was 0.74.

#### 2.3.3. Civic Discussions with Parents

Adolescents were administered a scale adapted by Flanagan et al. [60]. It consists of three items: “I speak to my parents about issues affecting our community”; “I am interested in my parents’ views on issues affecting our community”; and “My parents encourage me to express my views on issues affecting our community, even if they are different from their views”. Items were rated on a 5-point scale ranging from 1 (*strongly disagree*) to 5 (*strongly agree*). To test the factorial validity of the scale, we conducted a CFA based on the ML estimation procedure. The model was saturated; therefore, its fit to the data was not tested, but factor loadings were high and significant, ranging from 0.65 to 0.87. The internal consistency of the scale was good: factor determinacy was 0.92, and Cronbach’s α was 0.81.

### 2.4. Data Analyses

As a preliminary step, we performed descriptive statistics for the study variables, as well as Pearson correlations. We also conducted a series of Multivariate Analyses of Variance (MANOVA) to investigate whether gender differences in the study variables exist. To test the hypothesized model, a structural equation model (SEM) with latent variables was run. Adolescents’ gender (1 = male; 2 = female) and age, and parents’ educational level were entered as covariates into the model. Because the study variables showed a multivariate, non-normal distribution (normalized Mardia’s coefficient = 52.96, *p* < 0.001), we used the robust ML estimation method, which adjusts standard errors of parameter estimates and χ^2^ statistics (SBχ^2^) to account for non-normality. The overall fit of each model was tested using several goodness-of-fit statistics and an evaluation of the appropriateness of the model parameters. The χ^2^ statistic assessed the implied covariance matrix compared with a good-fitting model indicated by a non-significant result. Given the potential limitation of the χ^2^ test (it should be non-significant with *p* > 0.05), due to its tendency to reject the null hypothesis with large sample sizes and complex models, we relied on well-known goodness-of-fit indices and their related cut-offs to evaluate model fit [61]: CFI ≥ 0.90 for acceptable and ≥0.95 for good fit, RMSEA ≤ 0.08 for acceptable and ≤0.05 for good fit, and SRMR ≤ 0.10 for acceptable and ≤0.05 for good fit.

After that, in order to test the potential moderating role of environmental sensitivity in the relation between civic discussions with parents and civic engagement, we used the latent moderated structural equation (LMS) method [62], as suggested by Maslowsky et al. [63]. Because the simultaneous specification of more than one interaction term enhances multicollinearity and its severe effects on parameter estimations [64,65], we considered a separate LMS model for each moderating variable (i.e., AES, LST, EOE, and global environmental score), and we followed three steps. First, we estimated a structural SEM including the measurement model of the latent variables and the structural paths from civic discussions with parents and the moderating variable (i.e., AES, LST, EOE, and total environmental sensitivity score) to civic attitudes and civic behaviours without the latent interaction term *civic discussions × environmental sensitivity* (Model 0). Second, we entered this latent interaction into the model (obtaining the LMS model) to predict civic attitudes and estimated it (Model 1). Third, we included the latent interaction term for predicting civic behaviours and estimated it (Model 2). The output of Model 2 reported the final standardized regression coefficients (we obtained standardized beta coefficients by standardizing the data prior to analyses) and showed whether the latent interaction was significant. If significant, the interaction was interpreted by graphing as in standard regression models [66]. The model fit indices (i.e., χ^2^, CFI, RMSEA, and SRMR) and their associated cut-offs for Model 0 were as mentioned above. However, no model fit indices have been developed for the LMS models (Model 1 in this study). Alternatively, following the suggestions provided by Klein and Moosbrugger [62] and by Muthen [67], we compared the relative fit of Model 0 (where the interaction is not estimated and therefore assumed to be zero) and Models 1 and 2 (where the interaction is estimated) using a log-likelihood ratio test, used to determine whether the more parsimonious model represents a significant decrement in fit relative to the more complex model [68]. The test statistic for the log-likelihood ratio test, expressed as *D*, was calculated using the following equation: *D* = −2[(log-likelihood for Model 0) − (log-likelihood for Model 1)]. The values of *D* are almost distributed as χ^2^. We calculated the degrees of freedom (*df*) to determine the significance of *D* by subtracting the number of free parameters in the more parsimonious model from the number of free parameters in the more complex model.

## 3. Results

### 3.1. Descriptive Statistics and Correlations

Table 1 depicts the descriptives of each examined variable, as well as their bivariate correlations. Overall, the data showed a univariate distribution, with the skewness and kurtosis values falling in the range between −1.0 and +1.0. Civic attitudes and behaviours were positively and significantly associated with each other; they were positively and significantly related to AES and LST, as well as the environmental sensitivity total score, but unrelated to EOE; they were also positively and significantly related to civic discussions with parents. Civic discussions with parents were positively and significantly related to the environmental sensitivity total score and AES subscale.

### 3.2. Gender Differences in Study Variables

With regard to civic engagement, the results of MANOVA showed a significant multivariate effect of gender (Wilks’ λ = 0.94, *F*(2, 435) = 13.51, *p* < 0.001; η^2^ = 0.06). Univariate ANOVAs indicated that females reported higher levels of civic attitudes than males, but the effects were small. With regard to environmental sensitivity, the results of the MANOVA showed a significant multivariate effect of gender (Wilks’ λ = 0.95, *F*(3, 434) = 7.83, *p* < 0.001; η^2^ = 0.05). Univariate ANOVAs indicated that females reported higher levels on all dimensions than males, but the effects were small. With regard to civic discussions with parents, the results of ANOVA showed a significant effect of gender (*F*(1, 436) = 10.70, *p* < 0.001; η^2^ = 0.05), with females reporting higher levels than males, but the effect was small. The results are reported in Table 2.

### 3.3. Relations of Civic Engagement with Environmental Sensitivity and Civic Discussions with Parents

The hypothesized model had a good fit to the data, SBχ^2^ (215) = 429.91, *p* < 0.001, CFI = 0.904, RMSEA = 0.048, and SRMR = 0.043. The standardized solution is reported in Figure 1. With regard to civic discussions with parents, they were positively and significantly associated with only one dimension of environmental sensitivity, that is, AES. They were also positively and significantly related to both civic attitudes and civic behaviours. Finally, with regard to environmental sensitivity dimensions, EOE was negatively and significantly related to civic behaviours only, and AES was positively and significantly related to civic attitudes only; LST was not significantly related to civic attitudes or behaviours.

We then tested the hypothesized pattern of associations considering the environmental sensitivity total score. The hypothesized model showed the following fit indices, SBχ^2^ (82) = 226.90, *p* < 0.001, CFI = 0.904, RMSEA = 0.064, and SRMR = 0.041. The standardized solution is depicted in Figure 2. Environmental sensitivity and civic discussions with parents were positive associated with each other. Civic discussions with parents significantly predicted both civic attitudes and behaviours, whereas environmental sensitivity resulted as a significant predictor of civic attitudes only.

### 3.4. Relations of Civic Engagement with Civic Discussions with Parents: The Moderating Role of Environmental Sensitivity

The goodness-of-fit indexes of models testing the moderating role of environmental sensitivity on the relation between civic engagement and civic discussions with parents are summarized in Table 3. We ran a series of models for the environmental sensitivity total score, as well as the three subdimensions (i.e., EOE, AES, and LST).

#### 3.4.1. Interaction of Civic Discussions with Parents and EOE Predicting Civic Engagement

Model 0 (including the measurement model of the latent variables and the structural paths from discussions with parents and EOE to civic attitudes and civic behaviours, without the latent interaction terms) showed an adequate fit, χ^2^(82) = 143.02, *p* < 0.001, CFI = 0.956, RMSEA = 0.041, and SRMR = 0.042. Only discussions with parents significantly predicted civic attitudes and civic behaviours. We then tested Model 1 (including the latent interaction *civic discussions × EOE* for civic attitudes). The relative fit of Model 1 versus Model 0 was determined via a log-likelihood ratio test comparing the log-likelihood values of Model 0 and Model 1, producing a log-likelihood difference value of *D* = 1.64. Considering the number of free parameters of Model 0 (51) and Model 1 (52), the difference in free parameters was = 1, representing the *df* value to be used for the log-likelihood ratio test. Using a chi-square distribution, this log-likelihood ratio was not significant, suggesting that Model 0 (without the interaction effect) did not represent a significant loss in fit relative to the Model 1 (with the interaction effect). We then evaluated Model 2 (including the latent interaction *civic discussions × EOE* for both civic attitudes and civic behaviours). The relative fit of Model 2 versus Model 1 was determined via the log-likelihood ratio test comparing the log-likelihood values of Model 1 and Model 2, generating a log-likelihood difference value of *D* = 0.56. This log-likelihood ratio was not significant, suggesting that Model 1 (with the interaction effect for civic attitudes only) did not represent a significant loss in fit relative to Model 2 (with both interaction effects). Overall, the findings indicated that EOE did not moderate the associations between discussions with parents and civic attitudes and behaviours, and that only the main effects of discussions with parents on civic engagement were to be considered.

#### 3.4.2. Interaction of Civic Discussions with Parents and AES Predicting Civic Engagement

Model 0 reported adequate fit indices, χ^2^(69) = 175.05, *p* < 0.001, CFI = 0.918, RMSEA = 0.059, and SRMR = 0.078. Only discussions with parents significantly predicted civic attitudes and civic behaviours. We then tested Model 1. The relative fit of Model 1 versus Model 0 was determined via the log-likelihood ratio test; the test value (*D*(1) = 2.02, *p* > 0.05) was not significant, showing that Model 0 did not represent a significant loss in fit relative to the Model 1. We then evaluated Model 2. The relative fit of Model 2 versus Model 1 was determined via the log-likelihood ratio test; the test value (*D*(1) = 2.14, *p* > 0.05) was not significant, indicating that Model 1 did not show a significant loss in fit relative to Model 2. Thus, results showed that AES did not moderate the relations between discussions with parents and civic attitudes and behaviours, and that only the main effects of discussions with parents on civic engagement were to be considered.

#### 3.4.3. Interaction of Civic Discussions with Parents and LST Predicting Civic Engagement

Model 0 showed adequate fit indices, χ^2^(46) = 66.85, *p* = 0.02, CFI = 0.981, RMSEA = 0.032, and SRMR = 0.030. Only discussions with parents resulted as a significant predictor of civic attitudes and civic behaviours. Model 1 was then estimated. We determined the relative fit of Model 1 versus Model 0 via the log-likelihood ratio test; the test value (*D*(1) = 0.84, *p* > 0.05) was not significant, meaning that Model 0 did not represent a significant loss in fit relative to the Model 1. Model 2 was then estimated. The relative fit of Model 2 versus Model 1 was determined via the log-likelihood ratio test; the test value (*D*(1) = 0.02, *p* > 0.05) was not significant, indicating that Model 1 did not report a significant loss in fit relative to Model 2. Therefore, these findings indicated that LST did not moderate the associations between discussions with parents and civic attitudes and behaviours, and that only the main effects of discussions with parents on civic engagement were to be considered.

#### 3.4.4. Interaction of Civic Discussions with Parents and the Environmental Sensitivity Total Score Predicting Civic Engagement

Model 0 fit the data adequately, SBχ^2^ (82) = 226.90, *p* < 0.001, CFI = 0.904, RMSEA = 0.064, and SRMR = 0.041. We then tested Model 1. We determined the relative fit of Model 1 versus Model 0 via the log-likelihood ratio test; the test value (*D*(1) = 3.56, *p* > 0.05) was not significant, indicating that Model 0 did not show a significant loss in fit compared with the Model 1. We then tested Model 2. The relative fit of Model 2 versus Model 1 was determined via the log-likelihood ratio test; the test value (*D*(1) = 0.30, *p* > 0.05) was not significant, indicating that Model 1 did not represent a significant loss in fit relative to Model 2. Globally, the findings indicated that environmental sensitivity did not moderate the relations between discussions with parents and civic attitudes and behaviours, and that only the main effects of discussions with parents on civic engagement were to be taken into account.

## 4. Discussion

This study aimed to broaden the current literature on civic engagement in youth in order to evaluate which factors promote and enhance its development. In particular, we focused on the association between adolescents’ civic engagement with a dispositional characteristic (environmental sensitivity) and a contextual factor (discussions with parents about civic issues), and their interplay. The present findings partially confirmed our hypotheses.

We first investigated the association between environmental sensitivity and civic engagement, measured in terms of civic attitudes and behaviours. In line with some authors [34,35,36], we considered environmental sensitivity as a multidimensional construct composed of three distinct facets (AES, LST, and EOE) and the total score. We found different patterns of associations between these facets and both civic attitudes and behaviours.

Specifically, AES was the only dimension showing positive associations with civic attitudes. Most likely, AES—which captures aesthetic sensitivity (e.g., being deeply moved by arts and music)—positively influences the way in which individuals approach their surrounding environments, promoting a higher propensity toward the attention to others’ needs, improving the ability to efficiently cooperate with others, as well as enhancing greater interest in obtaining new information. In line with this claim, previous studies reported that AES was significantly associated with openness and agreeableness [36,39,40,69], highlighting that this dimension seems to capture sensitivity to more attractive, graceful, and harmonious aspects of the environment. However, contrary to our hypotheses, AES did not have any significant association with civic behaviours. As a viable explanation, even if attitudinal inclinations and behaviours usually work hand-in-hand, some authors reported a gap between the two dimensions, highlighting that what people think about an issue does not always coincide with the behaviour. As an example, studies conducted by Whitley and Yoder [70] and by Hwang and Yeo [71] examined discrepancies between attitudes and behaviours, evidencing that the subjective propensity toward an issue and the actual behaviour may not be perfectly aligned with each other. From this point of view, we suggest conducting additional research in order to further examine whether AES positively affects attitudes but not behaviours, as well as defining more specific civic behaviours.

Contrary to our hypotheses, EOE negatively predicted civic behaviours. EOE, which concerns the feeling of discomfort, irritation, and annoyance when people have to perform too many tasks at once, may influence individuals to prioritise more proximal activities and demands (e.g., homework). For example, although individuals higher in EOE may be interested in community issues, they may decide not to take part in volunteering activities or other forms of civic participation due to lack of time or energy. However, more research should be conducted to further examine this statement.

Additionally, contrary to our expectations, LST was not related to any of the two dimensions of civic engagement, meaning that the unpleasant sensory arousal to external stimuli (i.e., LST) does not affect the way individuals relate to community issues. After all, LST has been associated with negative external stimuli [34], and this may likely explain the lack of significant associations with positive beliefs about the commitment in the community, as well as the low involvement in civic activities. Additional studies are needed to better understand such assumption. Briefly, these results indicate the usefulness of taking into account the three dimensions of environmental sensitivity separately because each of them was related to civic attitudes and behaviours in a different manner.

When considering environmental sensitivity as a total score, our findings provided evidence that it significantly predicted civic attitudes, but not civic behaviours. The same explanations may apply as to the findings regarding AES. This may further suggest that AES captures the core aspects of environmental sensitivity (i.e., perceive and process information about the environment).

We then investigated the influence of civic discussions with parents on adolescents’ civic engagement. As expected, and in line with previous studies [17,21,22,23,26,27], talking with parents about community issues was a significant predictor of youths’ positive beliefs about their commitment in the community and higher participation in civic activities. From this point of view, our study provides further support to the importance parents have in shaping their children’s way of interacting with their surrounding environment. Parents who stimulate open communication at home may likely foster their children’s ability to engage and connect with others, also improving civic values. Discussing issues affecting one’s own community with parents may promote civic engagement because it involves the development and employment of civic skills, including expressing opinions, respecting others’ points of view, and being open-minded toward diversity [20,21]. On a practical level, our research highlights the need to encourage parents to include and promote civic topics within the discussions with offspring, thus underlining the relevance of family as a protective factor in predicting positive outcomes.

In the same vein, an inspection of the associations between sensitivity and civic discussions with parents showed that only AES was significantly and positively associated, which is in line with other research suggesting that AES correlates with positive stimuli [34]. Thus, AES seems to represent the facet of environmental sensitivity most strongly related to civic engagement. In light of this, we humbly claim that this specific dimension may be particularly relevant in the family context nurturing to the value of beauty (in its multiple and assorted forms) and to its protection and safeguard.

However, bear in mind that discussions with parents—as assessed in the current study—did not exclusively evaluate the extent to which parents were able to promote civic discussions within the family. Indeed, two out of the three items used for measuring this variable addressed the active contribution of youth in determining this contextual factor (i.e., “I speak to my parents about issues affecting our community”; “I am interested in my parents’ views on issues affecting our community”); therefore, these items seem to also tap adolescents’ will and inclination to know and talk with their parents about civic issues, as well as their actual behaviours in this domain. In light of the above considerations, our findings about the role of discussions with parents on children’s civic engagement should be interpreted with caution because they may be biased because of the specific used descriptors. In order to avoid this issue, future studies should include multiple sources of information.

As a second goal, we examined whether adolescents’ environmental sensitivity moderate the relation between civic discussions with their parents and their civic engagement. Contrary to our expectations, our analyses did not detect significant moderating effects of environmental sensitivity. To try to account for this unexpected result, it is necessary to jointly examine the previously discussed findings. In general, more developed civic attitudes seem to be associated with higher levels of aesthetic sensitivity and—at the same time—with a family context in which parents and children talk about the problems afflicting their own community; in turn, these two factors—the personal and the contextual—seem to reinforce each other. Together, these findings suggest the idea that growing up in a family attentive to the well-being of the community and capable of democratically accepting the points of view of its members could allow youths to develop a particular sensitivity for the beautiful and harmonious aspects of the environment in which they live, also stimulating their interest in the collective good and its care, as well as enhancing communication skills essential for social participation. However, the study failed to detect the hypothesized interaction effect between personal and contextual factors in youth civic engagement. As a viable explanation, the specific aspect of the family context considered in the present study can be an important factor for civic engagement, regardless of the personal characteristics of the young person. The kind of interactions that adolescents have with parents are of fundamental relevance because parents play a crucial role in shaping their children’s attitudes and behaviours throughout life, also representing an ecological “developmental asset” [72] affecting the positive development of youth, and in youth contributions to their communities [73,74]. As a further explanation, we postulated that these interaction effects are in line with the Vantage Sensitivity framework [43,44], according to which more sensitive individuals are more likely to benefit from positive environmental influences. Nevertheless, sufficient information about the nature of these civic discussions is lacking, and stating that it represents a positive environmental influence may be somewhat vague; indeed, except for one item that describes parents’ interest in understanding their children’s perspectives (which can be seen as a positive influence), we cannot exclude the likelihood of disagreements and arguments, which may in turn undermine the quality of the family atmosphere.

Finally, we tested gender differences in all the study variables. As expected, and in line with previous studies [34,48,49,52,54,55,57], females showed higher levels in all the examined variables, with the only exception of civic behaviours, which did not report any significant differences between the two genders. As noted by some authors [75,76], gender differences (or the lack of) may vary depending on the type of civic behaviour being evaluated; thus, additional research should assess different and more specific civic behaviours to further examine whether gender differences may exist. Notably, although we found gender differences, the estimated effect was small, suggesting the lack of a substantial and considerable difference between males and females. Thus, future studies are needed to extend the knowledge of gender differences in our examined variables.

### Limitations and Suggestions for Future Research

Our findings should be considered in light of several limitations. First, the cross-sectional nature of the work does not allow us to provide evidence of any causal relationships among the study variables. Longitudinal studies are needed and highly recommended, especially for the specific population examined, because they could offer a better and deeper comprehension of how civic engagement develops and changes across development, also ascertaining temporal ordering and causal relationships among the variables (through cross-lagged models). Second, we did not consider other relevant contextual predictors of civic engagement in youth, such as parents’ civic participation and involvement, or other individual characteristics, such as personal motivations and personality traits. Future studies should examine the influence of a larger number of predictors of civic engagement. Third, we used only self-report instruments, which might have produced distorted results due to shared variance and social desirability. Additional studies should also include qualitative techniques for a deeper understanding of the meaning adolescents attribute to civic engagement. Future research may address investigations into some of the questions that arose in the current study: (a) Is AES more susceptible to the influence of family context? If so, which factors promote its development? (b) What is the role of EOE in stimulating or inhibiting youth behaviour, especially in ambiguous contexts where there are not clear and well-defined rules that can help to regulate behaviour? (c) Can environmental sensitivity be a significant moderator in the relationship between parents’ civic engagement and youth’s civic engagement?

## 5. Conclusions

Despite these limitations, this study contributes to both research and practice related to the promotion of civic engagement in youth. Regarding the field of research, our study offers an advancement in the current literature on civic engagement among youth providing insights into previously unexplored variables that might be associated with it. Specifically, the significant effects of some dimensions of environmental sensitivity on both civic attitudes and behaviours indicate that the investigation of this dispositional trait should be useful to better understand the mechanisms underlying the development and occurrence of civic engagement among adolescents. In addition, our study suggests that it could be useful to take into account the three dimensions of environmental sensitivity separately because each of them had different associations with civic attitudes and behaviours. Therefore, each facet of environmental sensitivity, although related to each other, reflects a specific aspect of the construct, which may likely show a different pattern of associations with related variables. Additionally, because AES seems to describe the core element of environmental sensitivity, our study highlights that individuals who report a greater appreciation of aesthetic stimuli tend to show higher discussions with their parents, as well as a higher propensity toward community issues. Indeed, when examining the three sub-dimensions, only AES was significantly associated with civic attitudes and civic discussions, and the same pattern of relationships was estimated when the total score of environmental sensitivity was taken into account (it is plausible that the significant associations between sensitivity, civic discussions, and civic attitudes were mainly driven by the AES subscale). Being sensitive to aesthetic stimuli may be likely associated with a higher inclination toward searching for emotional bonds with people living in the surrounding environments, which represent the basis for the development of a sense of belonging and enhancing positive attitudes toward concerns related to the larger community. Moreover, the findings underline the need to jointly examine individual and dispositional predictors of youth civic engagement in order to obtain broader comprehension of this phenomenon.

With regard to the field of practice, our study emphasizes the importance of promoting interventions to address training and encouraging parents to include and promote civic topics within meaningful family discussions with their children. In this way, they can help children to develop their own views on political and civic issues, as well as to foster in them behaviours of active research of information on civic matters.

## Figures and Tables

**Figure 1 ijerph-20-06315-f001:**
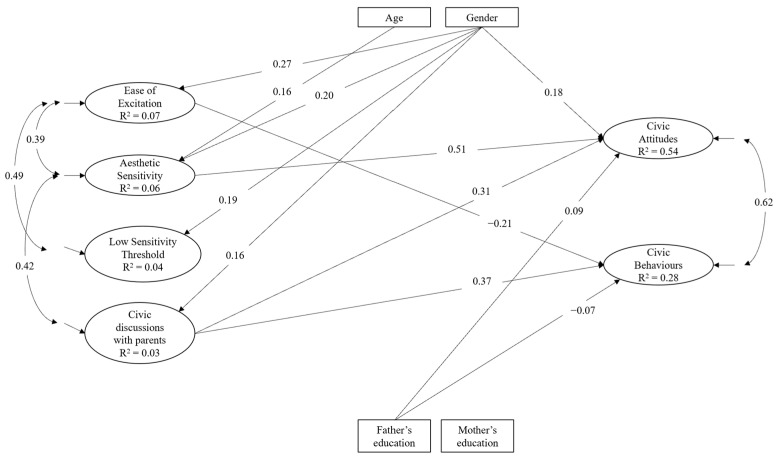
Statistical model of relations between civic engagement, environmental sensitivity (as single dimensions), and civic discussions with parents. Only significant (*p* < 0.05) paths and correlations are reported.

**Figure 2 ijerph-20-06315-f002:**
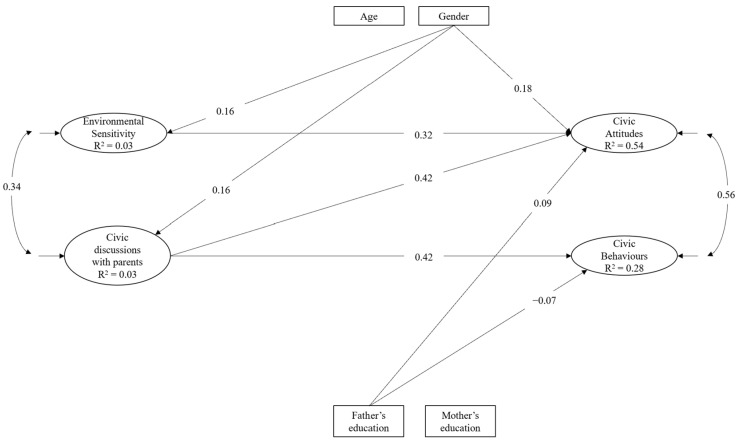
Statistical model of relations between civic engagement, environmental sensitivity (as global construct), and civic discussions with parents. Only significant (*p* < 0.05) paths and correlations are reported.

**Table 1 ijerph-20-06315-t001:** Descriptive statistics and Pearson correlation coefficients of study variables (*n* = 438).

Variable	1	2	3	4	5	6	7	8	9	10
1. CA	-									
2. CB	0.47 ***	-								
3. EOE	0.09 *	.02	-							
4. AES	0.35 ***	0.22 ***	0.23 ***	-						
5. LST	0.14 **	0.11 *	0.34 ***	0.15 **	-					
6. ES (Total score)	0.24 ***	0.13 **	0.85 ***	0.59 ***	0.63 ***	-				
7. Civic Discussions with Parents	0.40 ***	0.37 ***	0.01	0.28 ***	0.03	0.12 *	-			
8. Age	0.03	0.03	0.01	0.11 *	0.07	0.07	−0.01	-		
9. Father’s Education (years)	0.11 *	0.11 *	−0.06	−0.04	0.02	−0.05	0.05	−0.06	-	
10. Mother’s Education (years)	0.10 *	0.12 *	−0.06	−0.07	0.03	−0.05	0.06	−0.06	0.60 ***	-
*M*	3.88	2.83	3.36	4.23	3.27	3.66	3.76	16.50	2.70	2.83
*SD*	0.43	0.75	0.76	0.56	1.02	0.53	0.83	1.36	0.90	0.91
Skewness	−0.43	0.14	−0.19	−0.80	−0.06	−0.16	−0.97	−0.52	0.31	0.20
Kurtosis	0.66	−0.23	−0.02	0.43	−0.51	−0.13	1.40	−0.92	0.32	0.05

Note: CA = civic attitudes; CB = civic behaviours; EOE = Ease of Excitation; AES = Aesthetic Sensitivity; LST = Low Sensitivity Threshold; ES = environmental sensitivity. * *p* < 0.05. ** *p* < 0.01. *** *p* < 0.001.

**Table 2 ijerph-20-06315-t002:** Differences in mean scores of study variables between males (*n* = 131) and females (*n* = 307) and F tests.

	Males		Females		ANOVA			
	*M*	*SD*	*M*	*SD*	*F*	*df*	*p*	η^2^
CE								
CA	3.72	0.46	3.95	0.39	26.84	1, 436	<0.001	0.06
CB	2.72	0.81	2.87	0.71	3.82	1, 436	0.051	0.01
ES								
ES dimensions								
EOE	3.13	0.86	3.46	0.69	17.29	1, 436	<0.001	0.04
AES	4.12	0.58	4.27	0.55	6.81	1, 436	<0.001	0.02
LST	3.04	1.04	3.37	1.00	9.82	1, 436	<0.001	0.02
ES total score	3.47	0.57	3.74	0.49	23.52	1, 436	<0.001	0.05
Civic discussions with parents	3.56	0.85	3.84	0.80	10.70	1, 436	<0.001	0.02

Note: CE = civic engagement; CA = civic attitudes; CB = civic behaviours; ES = environmental sensitivity; EOE = Ease of Excitation; AES = Aesthetic Sensitivity; LST = Low Sensitivity Threshold.

**Table 3 ijerph-20-06315-t003:** Goodness-of-fit indexes of models testing the moderating effects of environmental sensitivity (both in its components, as well as a global construct).

	Log-Likelihood	n. Parameters	BIC	Model Comparison	*D*	*Ddf*
EOE						
M0 Null model	−7289.27	51	14,888.73	-	-	-
M1 Interaction for civic attitudes	−7288.45	52	14,893.17	M0-M1	1.64 ns	1
M2 Interaction for civic attitudes and behaviors	−7288.17	53	14,898.70	M1-M2	0.56 ns	1
AES						
M0 Null model	−6143.85	48	12,579.65	-	-	-
M1 Interaction for civic attitudes	−6142.84	49	12,583.70	M0-M1	2.02 ns	1
M2 Interaction for civic attitudes and behaviors	−6141.77	50	12,583.64	M1-M2	2.14 ns	1
LST						
M0 Null model	−5447.81	42	11,151.07	-	-	-
M1 Interaction for civic attitudes	−5447.39	43	11,156.32	M0-M1	0.84 ns	1
M2 Interaction for civic attitudes and behaviors	−5447.38	44	11,162.38	M1-M2	0.02 ns	1
ES						
M0 Null model	−5557.17	45	11,388.04	-	-	-
M1 Interaction for civic attitudes		46	11,390.56	M0-M1	3.56 ns	1
M2 Interaction for civic attitudes and behaviors		47	11,396.34	M1-M2	0.30 ns	1

Note: EOE = Ease of Excitation; AES = Aesthetic Sensitivity; LST = Low Sensitivity Threshold; ES = environmental sensitivity.

## Data Availability

The data presented in this study are available from the corresponding author upon request.

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
