# Peer review of "Individual Differences in Adolescents’ Civic Engagement: The Role of Civic Discussions with Parents and Environmental Sensitivity"

_ijerph, 2023, doi:10.3390/ijerph20136315_

Round 1

Reviewer 1 Report

This is an interesting paper examining the association among civic discussions with parents, environmental sensitivity and civic engagement, including both attitudes and behaviors.

The authors’ hypothesis is that civic discussions with parents, environmental sensitivity and their interaction affected both dimensions of civic engagement, accounting for gender and parents’ education level.

Results partially confirmed the authors’ hypotheses evidencing a main effect of civic discussions with parents on  both dimensions of civic engagement, and a main effect of environmental sensitivity on civic attitudes. The interaction between discussions with parents and environmental sensitivity had no effect on civic engagement. As concerns the role of specific dimensions of environmental sensitivity (Aesthetic Sensitivity, AES; Ease of Excitation, EOE; Low Sensitivity Threshold, LST), Aesthetic Sensitivity was found to be positively related to civic attitudes whereas Low Sensitivity Threshold was found to be negatively related to civic behaviors.

The main merit of the study is to expand our knowledge about the association between environmental sensitivity and civic engagement supporting the hypothesis that individuals with higher levels of sensitivity to external stimuli are more likely to be sensitive to civic issues.

The paper is well-written and clear, and analyses are correctly performed.

I believe that the paper has many merits for publication.

In my opinion, two critical points are present in the paper.  I do not think they affect the overall quality of the work, but I’d invite the authors to critically discuss these points when they  present the hypotheses and comment on the results.

The first point concerns considering “civic discussion with parents” as a contextual/environmental variable. I agree that a contextual variable is never totally external to individuals but in this case, this variable seems to describe an adolescent behavior or attitude more than a “contextual factor”. The active contribution of youth in determining this contextual variable seems prominent. Notably, two out of the three items of the scale start with the pronoun “I”: I speak to my parents about issues affecting our community”; “I am interested in my parents’ views on issues affecting our community.

While acknowledging the significant role parents play when their children show interest in discussing civic issues with them, which can be seen as a contextual factor, I believe it is crucial for the authors to recognize and discuss the potential bias arising from relying on a single source of information in this specific case.

A second point concerns the absence of interaction effects between ES and “discussion with parents” on civic engagement. Authors’ expectations are informed by the Vantage Sensitivity framework according to which more sensitive individuals are also more disposed to benefit from positive environmental influences. My doubt concerns the identification of “discussion with parents” in a positive environment. Actually, we do not have information regarding the nature of the discussions between youth and their parents, including the possibility of disagreements or the overall climate of these conversations. While one item on the scale addresses parents' interest in understanding their children's perspectives, doubts remain. It is possible that when children and parents engage in discussions about civic issues, it may also stem from disagreements between them. Consequently, the ambiguity in the construct could be a reason for the lack of interaction effects.

A critical comment concerning these two points in the discussion could critically enrich the discussion.

Minor points:

Line 41 – to define “recent” a paper by 2007 is not adequate.

Line 43 – Investigated: “I” is in a capital letter.

Line 54 – the sentence seems incomplete.

Line 55 – to define “discussion with parents” a situational factor seems excessive (see above)

Line 133 – “Unfortunately”(?). To be the first to investigate an issue is an exciting challenge for researchers.

Line 139 – it is better to not use causal language (“effect”)

Line 250 – spelling of “environmental”

Other points:

1) have the authors considered testing a model in which civic attitudes affect civic behaviors? Could it make sense?

2) Does the fact that only some subdimensions of ES affect Civic engagement offer new insights into our comprehension of the construct of environmental sensitivity?

Author Response

Dear Editor,

We are glad you appreciated our submitted manuscript and for the chance you offered us to resubmit a revised version of it.

We are thankful for all the recommendations provided. We think they were useful for improving the quality of our paper. 

In the present letter we copied the comments reported by the reviewers, and in italics we included the answers to each of them and the changes we made to our manuscript.

All changes were also added and highlighted (in red text) in the revised version of the paper that we attached.

In addition, we rephrased paragraphs containing overlaps with previous literature. They were highlighted in green.

Sincerely.

Reviewer 1

This is an interesting paper examining the association among civic discussions with parents, environmental sensitivity and civic engagement, including both attitudes and behaviors.

The authors’ hypothesis is that civic discussions with parents, environmental sensitivity and their interaction affected both dimensions of civic engagement, accounting for gender and parents’ education level.

Results partially confirmed the authors’ hypotheses evidencing a main effect of civic discussions with parents on both dimensions of civic engagement, and a main effect of environmental sensitivity on civic attitudes. The interaction between discussions with parents and environmental sensitivity had no effect on civic engagement. As concerns the role of specific dimensions of environmental sensitivity (Aesthetic Sensitivity, AES; Ease of Excitation, EOE; Low Sensitivity Threshold, LST), Aesthetic Sensitivity was found to be positively related to civic attitudes whereas Low Sensitivity Threshold was found to be negatively related to civic behaviors.

The main merit of the study is to expand our knowledge about the association between environmental sensitivity and civic engagement supporting the hypothesis that individuals with higher levels of sensitivity to external stimuli are more likely to be sensitive to civic issues.

The paper is well-written and clear, and analyses are correctly performed.

I believe that the paper has many merits for publication.

In my opinion, two critical points are present in the paper.  I do not think they affect the overall quality of the work, but I’d invite the authors to critically discuss these points when they present the hypotheses and comment on the results.

The first point concerns considering “civic discussion with parents” as a contextual/environmental variable. I agree that a contextual variable is never totally external to individuals but in this case, this variable seems to describe an adolescent behavior or attitude more than a “contextual factor”. The active contribution of youth in determining this contextual variable seems prominent. Notably, two out of the three items of the scale start with the pronoun “I”: I speak to my parents about issues affecting our community”; “I am interested in my parents’ views on issues affecting our community.

While acknowledging the significant role parents play when their children show interest in discussing civic issues with them, which can be seen as a contextual factor, I believe it is crucial for the authors to recognize and discuss the potential bias arising from relying on a single source of information in this specific case.

RESPONSE: Thanks for your comment. We totally agree with your suggestion, and we added the following paragraph in the revised version of the manuscript:

“However, it is noticeable to bear in mind that discussions with parents – as assessed in the current study - did not exclusively evaluated the extent to which parents were able to promote civic discussions within the family. Indeed, two out of the three items used for measuring this variable addressed the active contribution of youth in determining this contextual factor (i.e., “I speak to my parents about issues affecting our community”; “I am interested in my parents’ views on issues affecting our community”); therefore, these items seem to also tap adolescents’ will and inclination to know and talk about civic issues with their parents, as well as their actual behaviours in this domain. In light of the above considerations, our findings about the role of discussions with parents on children’s civic engagement should be interpreted with caution because they may be biased because of the specific descriptors used. In order to avoid this issue, future studies should include multiple sources of information”.

A second point concerns the absence of interaction effects between ES and “discussion with parents” on civic engagement. Authors’ expectations are informed by the Vantage Sensitivity framework according to which more sensitive individuals are also more disposed to benefit from positive environmental influences. My doubt concerns the identification of “discussion with parents” in a positive environment. Actually, we do not have information regarding the nature of the discussions between youth and their parents, including the possibility of disagreements or the overall climate of these conversations. While one item on the scale addresses parents' interest in understanding their children's perspectives, doubts remain. It is possible that when children and parents engage in discussions about civic issues, it may also stem from disagreements between them. Consequently, the ambiguity in the construct could be a reason for the lack of interaction effects.

RESPONSE: We are thankful for your precious comment. We perfectly agree with your suggestion, and we think that this issue may like to provide a further plausible explanation of the absence of interaction effects between ES and discussions with parents. We added the following paragraph in the Discussion section in the revised version of the manuscript:

“As a further explanation, we postulated these interaction effects in line with the Vantage Sensitivity framework [41,42], according to which more sensitive individuals are more likely to benefit from positive environmental influences. Nevertheless, sufficient information about the nature of these civic discussions is lacking, and stating that it represents a positive environmental influence may be somewhat doubtful and ambiguous; indeed, except for one item which describes parents' interest in understanding their children's perspectives (which can be seen as a positive influence), we cannot exclude the likelihood of disagreements and arguments, which may in turn undermine the quality of the family atmosphere”.

A critical comment concerning these two points in the discussion could critically enrich the discussion.

Minor points:

RESPONSE: We corrected the minor points in the revised version of the manuscript as follows:

Line 41 – to define “recent” a paper by 2007 is not adequate.

We removed the word “recent” 

Line 43 – Investigated: “I” is in a capital letter.

We changed “Investigated” into “investigated”

Line 54 – the sentence seems incomplete.

We are not sure whether we the sentence you think seems incomplete is “Thus, the current study is aimed at examining how contextual factors, such as discussions with parents about community issues, and dispositional factors, such as environmental sensitivity, are associated with adolescents’ civic engagement”.

If so, in our opinion it is a complete and meaningful sentence. 

Line 55 – to define “discussion with parents” a situational factor seems excessive (see above)

As said above, we agree with your comment about the limits of our measure of “discussion with parents”. However, from a theoretical point of view, in line with the literature on this topic (i.e., Wray-Lake & Schubert, 2019) , “discussions with parents” was defined as a contextual variable, then highlighting its limits in the Discussion, as you suggested,

Line 133 – “Unfortunately”(?). To be the first to investigate an issue is an exciting challenge for researchers.

We removed the word “Unfortunately”

Line 139 – it is better to not use causal language (“effect”)

We changed “effect” into “influence”

Line 250 – spelling of “environmental”

We corrected the misspelling of “environmetal” into “environmental”

Other points:

1) have the authors considered testing a model in which civic attitudes affect civic behaviors? Could it make sense?

RESPONSE: Thanks for your question. We think it is a relevant and interesting topic to be addressed. However, we did not test such a relationship because it was not the focus of the present paper.

In the submitted manuscript, we referred to civic attitudes and behaviors as two distinct dimensions of civic engagement, as also reported in previous studies (e.g., Cohen et al., 2013; Dassoneville et al., 2012; Di Marco et al., 2022).

Cohen, A. K., & Chaffee, B. W. (2013). The relationship between adolescents’ civic knowledge, civic attitude, and civic behavior and their self-reported future likelihood of voting. Education, citizenship and social justice, 8(1), 43-57.

Dassonneville, R., Quintelier, E., Hooghe, M., & Claes, E. (2012). The relation between civic education and political attitudes and behavior: A two-year panel study among Belgian late adolescents. Applied Developmental Science, 16(3), 140-150.

Di Marco, G., Hichy, Z., & Sciacca, F. (2022). Attitudes towards lockdown, trust in institutions, and civic engagement: A study on Sicilians during the coronavirus lockdown. Journal of Public Affairs, 22, e2739.

2) Does the fact that only some subdimensions of ES affect Civic engagement offer new insights into our comprehension of the construct of environmental sensitivity?

RESPONSE: Thanks for your question. As we reported in the Conclusions, we humbly point out the relevance of taking into account the specific subdimensions of environmental sensitivity because they showed a different pattern of associations with some related variables. To make this aspect clearer, we added the following paragraph in the main text:

“Therefore, each facet of environmental sensitivity, although related with each other, reflects a specific aspect of the construct, which may likely show a different pattern of associations with related variables”.

Reviewer 2 Report

IJERPH] Manuscript ID: ijerph-2449663

Individual differences in adolescents’ civic engagement: The role of civic discussions with parents and environmental sensitivity

This is a clearly written manuscript on a well executed study with a sample of Italian adolescents attending a high-school in Palermo, Italy. The questions addressed in the study add to the literature on civic engagement and especially the impact of civic discussions with parents on adolescents’ civic engagement. In general, the measures seem to be tapping relevant underlying constructs. Civic engagement is tapped by attitudes and behaviors and each dimension is modeled as a separate dependent variable. In terms of predictors, adolescents’ reports of their civic discussions with parents seems to be tapping the underlying construct – as indicated by the Cronbach’s alpha of .81 and the factor loadings on the scale.

My main concern was with the Environmental Sensitivity scale. First, the three sub-scales seem to tap very different aspects of a highly sensitive child; this is especially the case for Aesthetic Sensitivity (AES), somewhat less so for the Ease of Excitation (EOE) and Low Sensitivity Threshold (LST), the latter two tapping into a child’s being bothered or overwhelmed by environmental noise, distractions, or demands.  The alphas for the three sub-scales don’t engender a lot of confidence in their internal consistency. The full scale that is used in the models also has only a moderate internal consistency, .74. So one has to wonder what underlying construct is actually being measured – making it difficult to interpret the role of this variable for the outcome of civic engagement.

Second, the name for this predictor in the models (Environmental Sensitivity) is different from the original name for the scale (Highly Sensitive Child Scale) and is a bit misleading in the results and discussion. The original name for the 12-item scale seems more appropriate to the construct that the 12 items are tapping. In contrast, “Environmental Sensitivity” could mislead readers to think that the scale is tapping people’s (in this case adolescents’) sensitivity to the natural environment (similar to Environmental Identity scales).

Third, the results (Figure 2) are a bit misleading once the dimensions of environmental sensitivity are unpacked. One has to wonder whether the relationship (Fig. 2) of environmental sensitivity with civic attitudes (as well as the relationship of ES with civic discussions with parents) is driven mostly by AEA, in other words, adolescents who self-report higher appreciation of aesthetic stimuli also are more likely to report civic discussions with their parents and that they feel greater responsibility for their community (civic attitudes). Put succinctly, the results (especially in light of the fact that they are measured based on self-reports) may be picking up on the association of adolescents’ positive aesthetic sensitivities with their inclinations to emotionally bond and form affective relationships with others in their proximal (discussions with parents) and distal (community) environments. Developmentally, a sense of civic responsibility is nurtured by feelings of belonging to and appreciation of the fact that one is part of a larger community.

In the introduction, the authors note that their intention was to assess the potential (and understudied) contribution of dispositional factors to civic engagement. If anything, sensitivity to aesthetics is the dispositional factor for discussion. The hypotheses for moderator relationships was not well articulated in the introduction. So it wasn’t clear why they ran those analyses and, ultimately, there were no moderating effects of any ES dimension on the relationships between civic discussions with parents and civic engagement outcomes. I would recommend that the authors either lay out stronger arguments for testing each ES dimension as a moderator and/or  delete those analyses and just succinctly report that moderator models were tested but yielded no results.

Author Response

Dear Editor,

We are glad you appreciated our submitted manuscript and for the chance you offered us to resubmit a revised version of it.

We are thankful for all the recommendations provided. We think they were useful for improving the quality of our paper. 

In the present letter we copied the comments reported by the reviewers, and in italics we included the answers to each of them and the changes we made to our manuscript.

All changes were also added and highlighted (in red text) in the revised version of the paper that we attached.

In addition, we rephrased paragraphs containing overlaps with previous literature. They were highlighted in green.

Sincerely.

Reviewer 2

Individual differences in adolescents’ civic engagement: The role of civic discussions with parents and environmental sensitivity

 This is a clearly written manuscript on a well executed study with a sample of Italian adolescents attending a high-school in Palermo, Italy. The questions addressed in the study add to the literature on civic engagement and especially the impact of civic discussions with parents on adolescents’ civic engagement. In general, the measures seem to be tapping relevant underlying constructs. Civic engagement is tapped by attitudes and behaviors and each dimension is modeled as a separate dependent variable. In terms of predictors, adolescents’ reports of their civic discussions with parents seems to be tapping the underlying construct – as indicated by the Cronbach’s alpha of .81 and the factor loadings on the scale.

 My main concern was with the Environmental Sensitivity scale. First, the three sub-scales seem to tap very different aspects of a highly sensitive child; this is especially the case for Aesthetic Sensitivity (AES), somewhat less so for the Ease of Excitation (EOE) and Low Sensitivity Threshold (LST), the latter two tapping into a child’s being bothered or overwhelmed by environmental noise, distractions, or demands.  The alphas for the three sub-scales don’t engender a lot of confidence in their internal consistency. The full scale that is used in the models also has only a moderate internal consistency, .74. So one has to wonder what underlying construct is actually being measured – making it difficult to interpret the role of this variable for the outcome of civic engagement.

RESPONSE: Thanks for your comment.

We are aware that both the global scale and the three subscales showed moderate levels of internal consistency, indicating a somewhat doubtful items’ ability in describing the same underlying construct. This is particularly evident for the subscales but it may be reasonable due to the low number of items. Also previous studies (e.g., Nocentini et al., 2018; Pluess et al., 2018; Scrimm et al., 2018) reported similar results of Cronbach’s alpha for both the total scale, as well as for each subscale, even if these values are considered as acceptable.  For instance, see:

Bröhl, A. S., Van Leeuwen, K., Pluess, M., De Fruyt, F., Van Hoof, E., Weyn, S., & Bijttebier, P. (2022). Personality profile of the self-identified highly sensitive person: A lay theory approach. Journal of Individual Differences, 43(2), 95.

Lionetti, F., Aron, A., Aron, E. N., Burns, G. L., Jagiellowicz, J., & Pluess, M. (2018). Dandelions, tulips and orchids: Evidence for the existence of low-sensitive, medium-sensitive and high-sensitive individuals. Translational psychiatry, 8(1), 1-11.

Nocentini, A., Menesini, E., & Pluess, M. (2018). The personality trait of environmental sensitivity predicts children’s positive response to school-based antibullying intervention. Clinical Psychological Science, 6(6), 848-859.

Pluess, M., Assary, E., Lionetti, F., Lester, K. J., Krapohl, E., Aron, E. N., & Aron, A. (2018). Environmental sensitivity in children: Development of the Highly Sensitive Child Scale and identification of sensitivity groups. Developmental Psychology, 54(1), 51.

Scrimin, S., Osler, G., Pozzoli, T., & Moscardino, U. (2018). Early adversities, family support, and child well‐being: The moderating role of environmental sensitivity. Child: care, health and development, 44(6), 885-891.

Weyn, S., Van Leeuwen, K., Pluess, M., Lionetti, F., Greven, C. U., Goossens, L., Colpin, H., Van Den Noortgate, W., Verschueren, K., & Bastin, M. (2021). Psychometric properties of the Highly Sensitive Child scale across developmental stage, gender, and country. Current Psychology, 40(7), 3309-3325.

Although main literature considers ES as a global and unitary construct, there are some studies which point out that the three factors reflect different and specific facets, reporting specific patterns of associations with some related variables (Bröhl et al., 2022; Lionetti et al., 2018; Pluess et al., 2018; Weyn et al., 2021). Thus, in line with these assumptions, we aimed at examining whether they were differently associated with our study variables.  

Second, the name for this predictor in the models (Environmental Sensitivity) is different from the original name for the scale (Highly Sensitive Child Scale) and is a bit misleading in the results and discussion. The original name for the 12-item scale seems more appropriate to the construct that the 12 items are tapping. In contrast, “Environmental Sensitivity” could mislead readers to think that the scale is tapping people’s (in this case adolescents’) sensitivity to the natural environment (similar to Environmental Identity scales).

RESPONSE: We really appreciate your comment, and we agree with your opinion that readers non familiar with this research area may be confused regarding the terminology used. Nevertheless, throughout the manuscript, we clearly described the theoretical framework of ES, together with its conceptualization and definition, leaving small room to ES interpretations different to those reported. However, to make clearer that the HSCS is the scale used for evaluating ES, we added the sentence reported below in the revised version of the manuscript (Method section):

Adolescents were administered the Highly Sensitive Child Scale (HSC; [34], Italian version by Nocentini et al. [45], the most popular scale developed for assessing environmental sensitivity.”   

Third, the results (Figure 2) are a bit misleading once the dimensions of environmental sensitivity are unpacked. One has to wonder whether the relationship (Fig. 2) of environmental sensitivity with civic attitudes (as well as the relationship of ES with civic discussions with parents) is driven mostly by AEA, in other words, adolescents who self-report higher appreciation of aesthetic stimuli also are more likely to report civic discussions with their parents and that they feel greater responsibility for their community (civic attitudes). Put succinctly, the results (especially in light of the fact that they are measured based on self-reports) may be picking up on the association of adolescents’ positive aesthetic sensitivities with their inclinations to emotionally bond and form affective relationships with others in their proximal (discussions with parents) and distal (community) environments. Developmentally, a sense of civic responsibility is nurtured by feelings of belonging to and appreciation of the fact that one is part of a larger community.

 RESPONSE: Thanks for your accurate suggestion. We think it is very useful for providing further clarity to our results. We addressed this issue adding the following paragraph in the revised version of the manuscript (Conclusions section):

Additionally - because AES seems to describe the core elements of global environmental sensitivity - our study highlights that individuals who report a greater appreciation of aesthetic stimuli tend to show higher discussions with their parents, as well as a higher propensity toward community issues. Indeed, when examining the three sub-dimensions, only AES was significantly associated with civic attitudes and civic discussions, and the same pattern of relationships was estimated when global environmental sensitivity was taken into account (it is plausible that the significant associations between global sensitivity, civic discussions, and civic attitudes were mainly driven by AES subscale). Being sensitive to aesthetic stimuli may be likely associated with a higher inclination of searching for emotional bonds with people living in the surrounding environments, which represent the basis for the development of a sense of belonging, also enhancing positive attitudes toward concerns related to the larger community”. 

In the introduction, the authors note that their intention was to assess the potential (and understudied) contribution of dispositional factors to civic engagement. If anything, sensitivity to aesthetics is the dispositional factor for discussion. The hypotheses for moderator relationships was not well articulated in the introduction. So it wasn’t clear why they ran those analyses and, ultimately, there were no moderating effects of any ES dimension on the relationships between civic discussions with parents and civic engagement outcomes. I would recommend that the authors either lay out stronger arguments for testing each ES dimension as a moderator and/or delete those analyses and just succinctly report that moderator models were tested but yielded no results.

 RESPONSE: In order to make clearer our decision to test the moderator role of environmental sensitivity (as well as of its subdimensions) in the relationship between civic discussions and civic engagement, we added the following paragraphs in the Introduction section:

“Previous studies reported that environmental sensitivity was directly associated  with several psychological indicators of well-being [41,42], as well as it functioned as a significant moderator. Specifically, with regard to the moderating role, the Vantage Sensitivity framework [43,44] points out that [….]These studies provided some empirical evidence that environmental sensitivity moderates the effects of interventions, treatments, and parenting practices on children and adolescents’ psychological adaptation [44-47].”  

Previous studies reported that individuals tend to respond in a different manner to external stimuli as a function of their different levels of environmental sensitivity [45,56]. Although these studies considered environmental sensitivity as a global construct, we also investigated whether the three subdimensions functioned as significant moderators. Notwithstanding we did not have specific hypotheses regarding the inclusion of each specific facet in the moderation models, we explored such associations to provide addition knowledge in this specific research field”.

In addition, although moderation analyses did not show significant results, we think that their removal may weaken the global quality of the submitted paper. For this reason, we opted not to delete them in the revised version of the manuscript.